# The Serum Cell-Free microRNA Expression Profile in MCTD, SLE, SSc, and RA Patients

**DOI:** 10.3390/jcm9010161

**Published:** 2020-01-07

**Authors:** Barbara Stypinska, Anna Wajda, Ewa Walczuk, Marzena Olesinska, Aleksandra Lewandowska, Marcela Walczyk, Agnieszka Paradowska-Gorycka

**Affiliations:** 1Department of Molecular Biology, National Institute of Geriatrics, Rheumatology, and Rehabilitation, 02-637 Warsaw, Poland; 2Department of Connective Tissue Diseases, National Institute of Geriatrics, Rheumatology, and Rehabilitation, 02-637 Warsaw, Poland

**Keywords:** microRNA, miRNA serum levels, MCTD, ACTD

## Abstract

Mixed connective tissue disease (MCTD) is a rare disorder characterized by symptoms that overlap two or more Autoimmune Connective Tissue Diseases (ACTDs). The aim of this study was to determine whether miRNAs participating in the TLRs signaling pathway could serve as biomarkers differentiating MCTD or other ACTD entities from a healthy control group and between groups of patients. Although the selected miRNA expression level was not significantly different between MCTD and control, we observed that miR-126 distinguishes MCTD patients from all other ACTD groups. The expression level of miRNAs was significantly higher in the serum of systemic lupus erythematosus (SLE) and rheumatoid arthritis (RA) patients compared to controls. The miR-145 and -181a levels distinguished RA from other ACDT patients. miR-155 was specific for SLE patients. MiR-132, miR-143, and miR-29a distinguished RA and SLE patients from the systemic sclerosis (SSc) group. Additionally, some clinical parameters were significantly related to the miRNA expression profile in the SLE group. SLE and RA are characterized by a specific serum expression profile of the microRNAs associated with the Toll-like receptors (TLRs) signaling pathway. The analysis showed that their level distinguishes these groups from the control and from other ACTD patients. The present study did not reveal a good biomarker for MCTD patients.

## 1. Introduction

Autoimmune Connective Tissue Diseases (ACTDs) are autoimmune diseases characterized by spontaneous stimulation of the immune system and the production of antibodies against its own components of the nucleus and cytoplasm (proteins or nucleic acids). Classic ACTD includes systemic lupus erythematosus (SLE), rheumatoid arthritis (RA), systemic sclerosis (SSc), Sjögren’s syndrome (SS), and mixed connective tissue disease (MCTD). ACTDs are often associated with difficulties in diagnosis. Circulating antibodies in the blood often help in the patient classification and are included in the criteria for the classification of these diseases [1]. The most common ACTD is rheumatoid arthritis (affecting 1/100 people), whereas one of the least frequent is MCTD. The general prevalence of MCTD is still unknown, but in Caucasians it has been estimated to be 10/100,000. MCTD is a chronic immunological disorder that is defined by the presence of serum U1 antibodies of snRNP and overlapping features of at least two systemic diseases. There are many controversies regarding the definition and classification of MCTD. There are four sets of MCTD classification criteria and none of them is approved internationally, which further complicates its diagnostics. An important problem for clinicians is the clear distinction between MCTD and other ACTD, especially SLE and SSc. The criteria for classifying these units overlap in part. The classification is determined when the patient has a certain number of features/scores. It may happen that initially a disease is classified as an MCTD, but after further diagnosis, it turns out that it also meets the criteria of another ACTD. Some rheumatologists treat MCTD as a separate disease entity, others consider it only as a nonspecific stage of development of a separate ACTD [2]. Despite the controversy related to this disease, a long-term study on a large group of patients shows that in the majority of patients, the MCTD phenotype is stable [3]. In recent years, there has been significant progress in understanding the pathogenesis of MCTD and the central pathogenetic role of autoantibodies against the U1- small nuclear ribonucleoprotein antigen (anti-U1-RNP) autoantibodies has clearly emerged. These antibodies are present in almost all MCTD patients, but also in 30–40% of SLE patients [4,5]. In addition, anti-RNP can be observed during the onset of clinical symptoms. Long-term studies on MCTD patients prove that the presence of anti-U1 autoantibodies correlates with the activity of the disease, with the presence of the Raynaud’s phenomenon, and with lung injury. It was suggested that these autoantibodies may interact via Toll-like receptors (TLRs) signaling and that they are closely related to the production of IFN-1 [4].

MicroRNA (miRNA, miR) are short, noncoding RNAs that regulate gene expression on the posttranscriptional level and can influence an innate and adaptive inflammatory response. Some microRNAs are considered to be involved in the TLRs signaling during an inflammatory response. miR-181 may be a negative inflammatory regulator, as it inhibits TLR4 expression through direct binding to the 3′-UTR of this receptor. Expression of miR-181a reduces the production of TLR4 and pro-inflammatory cytokines interleukin (IL)-1β, IL-6, and tumor necrosis factor α (TNFα) [6]. Similarly to miR-181a, miR-143 negatively affects the expression of TLR2 and NF kappa beta. Liu et al.’s (2015) research indicates the involvement of miR-143 in the above mentioned process, however, the exact mechanism of its action is still unclear [7]. On the other hand, miR-155 inhibits the negative inflammatory regulators Src homology-2 domain-containing inositol 5-phosphatase 1 (SHIP1), suppressor of cytokine signaling 1 (SOCS1), EKKε, and TAB2, which leads to an increased inflammatory response. The production of this miRNA is induced by numerous TLRs ligands, which further proves its important role in regulating inflammatory response [8,9,10]. In addition, miR-155 regulates the distribution of T-cells. It has been shown that the absence of miR-155 reduces the level of IL4 and IL-17A in the serum (cytokines specific for Th2 and Th17 cells) [11]. Monocytes and macrophages show an increased expression of miR-132 in response to LPS. These microRNAs can act as negative regulators of signal transduction by lowering the expression of interleukin-1 receptor-associated kinase 4 (IRAK4) [10,12]. miR-126 is involved in many processes that lead to the pathogenesis of autoimmune diseases. However, the data on its expression and mechanism of action are not entirely clear [13]. miR-126, by targeting Tumor necrosis factor receptor (TNFR)-associated factor 6 (TRAF6), may inhibit the release of proinflammatory cytokines, which demonstrates its role in TLR signaling [14]. Liu et al. (2015) demonstrated the association of this miRNA with interferon (IFN) production. Low expression of miR-126 in the plasma of patients with SLE may cause an increase in IFN levels, thereby contributing to the development of this disease [15]. MiR-29a also contributes to inhibiting the response to intracellular bacterial infections and has been shown to target IFN-gamma mRNA directly [16]. All these data indicate that the microRNAs miR-155, -29a, -181a, -143, -145, -126, and -132 are involved in the regulation of the signaling pathway associated with TLRs activation and the production of pro-inflammatory cytokines.

Circulating miRNAs have many of the essential characteristics of good biomarkers such as stability (resistance to RNAses digestion, extreme pH, high temperatures, extended storage, and multiple freeze-thaw cycles) or easy availability [17,18]. miRNA may play a significant role in clinical applications, especially for the diagnosis of specific diseases, to monitor disease progression or the therapeutic effect of a drug on a patient [19]. MCTD like other ACTDs, may be associated with changes in the expression profile of circulating miRNAs, but there is still a paucity of data on the subject in the literature. It seems to us that determining the profile of circulating miRNAs, especially microRNAs that can play a role in the U1-RNP/TLRs/IFNs signaling pathway, for this unit is important and can have diagnostic significance. 

### Key Message

The number of serum circulating microRNAs that are important in the U1-RNP/TLRs/IFNs signaling pathway, including miR-126 miR-132 miR-143 miR-145 miR-155 miR-29a, miR-181a is distinctive, especially for SLE and RA groups of patients. MiR-29a, miR-145, miR-132, and miR-181a alone and the combination of the two microRNAs miR-145 and miR-181a have diagnostic value for RA distinction. Single microRNA expression was not enough to establish a significant SLE distinctive model, although combining the four miRNAs miR-126, miR-145, miR-29a, and miR-155 could allow us to create a statistically significant distinction model. The expression level of these microRNAs distinguished SLE and RA patients not only from the control group but also from other groups of patients suffering from ACTD. The expression of the investigated microRNAs did not differ between MCTD patients and the control group. However, the expression serum level of miR-126 distinguished MCTD patients from all other ACTD groups.

## 2. Materials and Methods

### 2.1. Patients and Clinical Characteristics 

Peripheral blood was obtained from 15 MCTD, 24 SLE, 37 SSc, 18 RA patients, and 75 healthy subjects without a history of ACTD. All patients were diagnosed at the Clinic and Polyclinic of Connective Tissue Diseases of the National Institute of Geriatrics, Rheumatology, and Rehabilitation in Warsaw. RA, SLE, and SSc patients fulfilled the revised EULAR/American College of Rheumatology (ACR) diagnostic criteria. MCTD patients were diagnosed in accordance with the diagnostic criteria of Kasukawa and/or Alarcon-Segovia and Villarreal. A clinical description of all patients is shown in Table 1. All participants provided informed written consent for participation in the study. The study was approved by the ethics committee in the National Institute of Geriatrics Rheumatology and Rehabilitation in Warsaw, Poland.

### 2.2. microRNA Relative Expression

microRNA from 200 μL freshly isolated serum was extracted using TRIzol Reagent (Invitrogen, Carlsbad, CA, USA) and AA Biotech MicroRNA Concentrator (A&A Biotechnology, Gdynia, Poland) according to the manufacturer’s protocol. Multiplexed Reverse Transcription and PreAmplification reactions were performed according to Protocol for Creating Custom RT and Preamplification (Publication Part Number 4465407 Revision Date January 2013 (Rev. C), Applied Biosystems by Life Technologies). The microRNAs expression level was determined using a real-time quantitative polymerase chain reaction (QRT-PCR), Taqman miRNA assays, and a TaqMan Fast Universal PCR Master Mix no AmpErase UNG (2X) (Applied Biosystems, Foster, CA, USA). Each target was measured in triplicate and normalized to the level of U6. The relative expression level of miRNAs was computed using the 2^−ΔΔCt^ method (Livak) with normalization to the endogenous small RNA control, U6. As a reference, we used results from a group of healthy people without any ACTD disease history.

### 2.3. Statistical Analysis

Differences between expression levels were determined using the Mann Whitney test, where a *p*-value < 0.05 indicated a statistically significant result. The receiver operating characteristic (ROC) curves analysis, the area under curves (AUCs), the likelihood ratio chi-square, and the *p* value obtained by multivariable logistic regression analysis were calculated using the R program. The cutoff value was estimated using a method that maximizes sensitivity and specificity simultaneously. The relation between expression and clinical parameters were measured using a Spearman’s correlation test and linear regression analysis. The statistical analysis and graph construction were carried out using the R program (R Development Core Team (2008) R: A language and environment for statistical computing. R Foundation for Statistical Computing, Vienna, Austria ISBN 3-900051-07-0, URL http://www.R-project.org.) with FSA [20], ggplot [21], ggpubr [22], pROC [23] R packages, and GraphPad Prism (version 8.0.0 for Windows, http://www.graphpad.com).

## 3. Results

### 3.1. miRNA Expression Level of Circulating microRNA in ACTD Patients and Healthy Subjects

The present study has demonstrated the expression profile of six microRNAs: miR-155, -143, -126, -29a, -181a, and -132 in serum from patients with MCTD (*n* = 15), SLE (*n* = 24), SSc (*n* = 37), and RA (*n* = 18), as well as in healthy subjects (*n* = 75). In patients with RA and SLE, all examined miRNAs were upregulated in comparison to controls. The levels of miR-155 (*p* = 0.03), -126 (*p* = 0.01), -29a (*p* = 0.004), -181a (*p* < 0.0001), -145 (*p* < 0.0001) and -132 (*p* = 0.007) were significantly higher in the sera of RA patients than in healthy subjects. Levels of miR-155 (*p* < 0.0002), -126 (*p* = 0.009), -29a (*p* < 0.0001), -143 (*p* = 0.003), -181a (*p* = 0.0009), -132 (*p* = 0.02), and -145 (*p* = 0.03), were significantly higher in the serum of SLE patients compared to in the control group (Table 2). SSc and MCTD microRNAs expression profiles did not differ significantly from healthy controls (Table 2).

To verify whether candidate microRNAs could discriminate RA and SLE patients from healthy controls, we performed ROC curve analysis and calculated AUC to evaluate their diagnostic potential (Figure 1 and Figure 2). The highest AUC differentiating RA from HC patients was obtained for miR-145 followed by miR181a, miR132, and miR-29a (Figure 1A,B). Setting a cutoff value at the level = 1.204, for miR29a, = 1.925 for miR-145, 1.747 = for miR132 and 1.679 = for miR181 allowed us to receive the highest specificity and sensitivity values (Figure 1C). Those results show that these microRNAs have the diagnostic potential to differentiate RA from healthy controls. However, in case of SLE, the highest AUC was obtained for miR-29a followed by miR-143, miR155, and miR181a (Figure 2A,B), while the results were not statistically significant (Figure 2).

In the next step, we analyzed a combination of selected microRNAs in order to evaluate whether they increase the diagnostic accuracy of the potential biomarker. We conducted multivariable logistic regression analysis including all of the miRNAs studied. Crucial factors in multivariate logistic regression model for RA and SLE differentiation have been analyzed using the likelihood ratio test. The combination of four microRNAs (miR-145, miR-155, miR-29a, and miR126) in SLE and combination of the two microRNAs miR-145 and miR-29a in RA turned out to be significant to that model (Figure 3A,B)

We observed that in both cases, according to the AUC value, combining microRNAs increases the diagnostic potential of biomarkers, compared to that provided by single microRNAs analysis (Figure 3C,D).

### 3.2. Differences in Circulating microRNA Expression Level between ACTD Patients

Next, we assessed whether the expression of examined microRNAs distinguishes ACTD patients from each other. 

Our analysis revealed that miR-126 expression in MCTD patients is significantly downregulated in comparison to other ACTDs (Figure 4A). In contrast, the level of miR-126 was the highest in patients with RA. The miR-126 level in RA patients was significantly higher compared to in patients with SSc (Figure 4A; *p* < 0.05) and MCTD (Figure 4A; *p* < 0.01). Additionally, we observed significantly higher expression of miR-181a and miR-145 in RA patients in comparison to in other ACTD patients (Figure 4B,C). The miR-181a level also differed between SLE and SSc patients (Figure 4B; *p* = 0.002). The miR-155 serum expression level was upregulated in SLE patients compared to other ACTD groups, but in relation to RA patients, this difference was not significant (Figure 4D; *p* = 0.054). Our analysis also showed that the levels of miR-132, miR-29a, and miR143 in serum were higher in SLE compared to in SSc patients (*p* = 0.007, *p* < 0.001, *p* = 0.002, respectively). Moreover, miR-132 and miR-29a were found at significantly lower levels in the sera of patients with SSc than in patients with RA (*p* = 0.01, *p* = 0.01, respectively).

### 3.3. Correlation between miRNAs Expression Levels and ACTDs Clinical Phenotype

Therefore, since significant differences in the levels of examined miRNA between study groups have been found, we decided to assess correlation analysis between expression levels of examined miRNAs and clinical parameters of ACTD patients. 

Relationships between the expression levels of selected microRNAs in serum and SLE activity were investigated using SLEDAI, SLICC, and CRP parameters. We noticed a slight inverse correlation between SLEDAI and expression levels of miR-155 (cor = −0.475, *p* = 0.06) and as well as between SLICC and expression levels of miR-132 (cor = −0.491, *p* = 0,06). Moreover, our analysis showed a significant association between examined miRNAs and an autoantibodies presence in SLE patients. Levels of miR-145 (cor = −0.579, *p* = 0.02) and miR-155 (cor = −0.608, *p* = 0.01) negatively correlated with the presence of anti-dsDNA antibodies (Appendix A). No association could be detected between levels of examined miRNAs and other clinical/serological features among patients with SLE.

The expression levels of microRNAs among patients with RA, MCTD, and SSc were not related to disease activity, the presence of antibodies, or specific clinical symptoms.

## 4. Discussion

ACTDs, and MCTD, in particular, are often associated with difficulties in diagnosis. Although circulating antibodies in the blood help in patient classification, obtaining a clear distinction between MCTD and other ACTD, especially SLE and SSc, is still a major challenge for clinicians. The recognition of the complexity of interactions between epigenetic mechanisms and immunity disorders in autoimmune diseases is essential for the research on fast and precise diagnosis as well as effective therapeutic strategies. To date, many microRNAs in serum have been identified as having the potential for good diagnostic biomarkers in ACTD. However, to our knowledge, no studies have been conducted on circulating microRNAs profile of MCTD patients. In the present study we observed the lowest level of miR-126 expression in MCTD patients, which distinguishes the MCTD group from other ACTD groups. Although its level was significantly downregulated in comparison to other ACTD patients, we have not observed statistically significant differences between MCTD patients and healthy subjects. miR-126 is present within the EGFL7 gene (epidermal growth factor like-domain 7) and according to Liakouli et al. (2019), it may be involved in the pathogenesis of SSc vasculopathy and fibrosis. miR-126 is a negative regulator of EGFL7 [24]. Other authors concluded that miR-126 may be involved in the initiation and development of systemic lupus erythematosus by inhibiting interferon production [15]. The present study revealed that miR-126 serum expression levels distinguished RA patients from SSc patients. A high serum level of miR-126 in RA patients has been revealed by Murata [25]. Although Wang et al. observed that miR-126 serum expression distinguishes SLE from RA patients in another study, we did not observe statistically significant differences [26]. Interestingly, in synovial fibroblasts, miR-126 does have an impact on the phosphoinositide 3-kinase/protein kinase B (PI3K/AKT) signaling pathway in RA patients, consequently SF apoptosis is blocked and proliferation is improved [25,27]. In SLE patients, it has been shown that over-expression of miR-126 is also associated with CD4+ overproduction and consequently with worsening of the patient state [28]. Other research revealed that the expression of miR-126 has been reported to be up-regulated in CD4+T cells from SLE and RA patients, leading to the demethylation of autoimmune-related genes and increased T cells activity and B cells stimulation [28,29]. Our previous studies on the analysis of the TLR/IFN signaling pathway and the miRNAs associated with it have shown that this pathway has the greatest pathogenic significance during the course of SLE. Based on our earlier research, we may observe that MCTD pathogenesis is completely different from the RA, SSc, or SLE pathogenesis, and these findings reflect differences in the inflammatory signatures of the MCTD patients. On the one hand, our results indicate that the miRNAs analyzed in the presented work are not characteristic of MCTD. Therefore, it is important to continue research in this area. But, on the other hand, our data highlight that the miRNA related to the U1-RNP/TLRs/IFNs signaling pathway may be another proof that MCTD can be distinct from other ACTD. Considering that this is the first microRNA study involving a group of people suffering from MCTD, it can only serve as a preliminary data for further analysis, and those results can be used in a meta-analysis that involves enough patients to draw correct conclusions.

Present study has shown that the serum expression levels of microRNAs significantly differentiates SLE and RA patients from other ACTD entities, as well as from healthy subjects. The expression level of all tested microRNAs (except for miR-143 in the RA group) was significantly higher in SLE and RA patients compared to in the healthy group. ROC curve analysis with an AUC greater than 0.7 confirmed the potential of four microRNAs: miR-29a, miR-145, miR181a, and –miR132, for RA differentiation. Cumulative logistic regression and the likelihood ratio test allowed for selection of two microRNAs: miR-145 and miR-29a, that together are crucial factors for RA detection. To date, several circulating microRNAs have also been indicated as potential diagnostic markers that differentiate between RA patients and healthy individuals. miR-125a-5p, mi-R24, and miR-26a (the pattern of expression of these three combined microRNA called as ePRAM together has greater specificity as a potential biomarker) [25], miR-15a-5p, 24-3p, -26a5p, -125a-5p, -146a-5p, -155-5p, -223-3p (here, similarly, the miR24, 26a, and 125a combination improved their parameters as an RA biomarker) [30], miRNA-4634 (miR-4634), miR-181d, and miR-4764-5p [31] in plasma were increased in RA patients compared to controls, whereas miR-342-3p, miR-3926, miR-3925-3p, miR-122-3p, miR-9-5p, and miR-219-2-3p [31] had a decreased level in RA patients compared to in healthy subjects. 

Earlier, studies on the expression profile of the circulating plasma microRNAs in SLE patients came to the conclusion that the level of miR-146a, miR-155 [26,32], miR-17, miR-20a, miR-106a, miR-92a, mi203 [33], miR-103, miR-150, miR-20a, miR-223, miR-27a, miR-15b, miR-16, miR-19b, miR-22, miR-23a, miR-25, miR-92a, and miR-93 [34] were lower, whereas the levels of miR-126, miR-21, miR-451, miR-223, miR-16 [26], miR-142-3p, and miR-181a [33] were higher in SLE patients than in healthy subjects. ACTDs are polygenic diseases with multiple sets of clinical manifestations in every individual case. Thus, the level of expression different miRNAs could be due to environmental factors, genetic factors, disease activity, the use of hormonal therapy, and even lifestyle.

Serum expression levels of miR-181a and miR-145 were significantly upregulated in RA patients compared to in other ACTDs. According to the literature, miR-181a distinguished SLE from HC but did not distinguish SLE from RA [34]. In the present study, the highest AUC was for miR-145 and miR-181 in RA patients. miR-145 plays a crucial role in the modulation of TNF-α mediated signaling and in cartilage matrix degradation. Chondrocytes stimulated by TNF-α downregulate miR-145 expression [35]. However in the case of miR-181a, it has been proved that it regulates DC immune inflammatory responses by targeting the JAK1-STAT1/C-Fos pathway [36]. We also observed that levels of miR-155 in the serum were specific for the group of SLE patients. Apart from healthy subjects, its expression in serum distinguished this group of patients from MCTD and SSc. MiR-155 expression levels were also upregulated in SLE compared to RA patients, but the difference was not significant. Gene miR-155 is located in the B cell integration cluster and originally was considered as a prooncogene associated with lymphoma. In 2015, Xin et al. showed that one of the miR-155 targets is gene S1pr1 (sphingosine-1-phosphate receptor 1), which might be involved in pathogenesis of SLE. According with the literature, SLE patients are characterized by a higher level of S1pr1, and Th17 from SLE patients also revealed a high level of S1pr1 mRNA. Moreover, miR-155–deficient mice are protected against the development of SLE lesions, hence this microRNA may be a useful therapeutic target in SLE treatment [11]. Studies show that miR-155 is implicated in other autoimmune diseases, such as RA and MS [37].

Some of circulating microRNA expression have shown a correlation with clinical factors in SLE patients, suggesting that they may be indicators of disease progression. We observed that th miR-155 expression level is inversely correlated with disease activity and the presence of anti-dsDNA antibodies. miR-132 expression levels were inversely correlated with disease activity in SLE patients. Wang et al (2010) demonstrated that the levels of miR-146a and miR-155 in the sera of SLE patients were associated with disease activity, proteinuria, the number of red blood cells, thrombocytes, and lymphocytes. After treatment with calcitriol for six months, the serum levels of miR-146a in SLE patients significantly increased and were also inversely correlated with calcium-phosphate products (r = −0.466, *p* = 0.003) [32]. In contrast, studies conducted up to now indicate that circulating microRNAs have the potential to be prognostic markers of disease activity for the occurrence of specific clinical symptoms or for the presence of specific autoantibodies for RA. For example, miR-4764-5p, miR-4634, miR-9-5p, miR-219-2-3p [31], miR-24, and ePRAM [25] exhibited significant correlations with either plasma cytokine and chemokine levels or clinical features for RA.

Due to the size of the sample and the clinical diversity of patients, our results have some limitations. Therefore, this data must be confirmed in larger studies. Circulating miRNAs have many features that make them good candidates for diagnostic or prognostic markers [18,38,39]. However, when interpreting the results and drawing conclusions, one should bear in mind the disadvantages of quantitative research on circulating miRNAs. The problem is the difficulty in obtaining repeatability of quantification of circulating microRNAs. It is caused by various assay platforms, varied methodology, protocols, sample preparation or isolation of miRNAs, as well as uniform and proper normalization. Differences in the results observed in the literature may also be related to ethnic differences or group validation. The large diversity between patients and the problem with methodological standardization indicate a problem involving using circulating microRNAs as diagnostic biomarkers. In addition, the mechanisms underlying aberrant circulating miRNAs involved in the pathogenesis of CTDs as well as the effects of other factors that regulate circulating miRNAs remain to be investigated. Functional experimental studies are required to verify and establish a causal relationship between improperly expressed circulating miRNAs and CTD.

## 5. Conclusions

Our research has shown that the levels of circulating microRNAs associated with the U1-RNP/TLRs/IFNs signaling pathway may be not a good diagnostic marker for MCTD in particular. Our studies provide a basis to look more closely at the importance of miR-126 serum expression that differentiated MCTD patients from other ACTD disease units. Moreover, we observed that the serum expression profile of these microRNA is characteristic for SLE and RA diseases entities. These microRNAs could serve as potential diagnostic biomarkers that differentiate them from healthy controls and/or from other ACTDs. Further research into the differences in the epigenetics of patients suffering from ACTD will allow a better understanding of the pathogenesis of these diseases, which in the future may lead to better diagnostics, differentiation of patients, particularly with regard to overlap syndromes, and appropriate classification of patients for treatment. 

## Figures and Tables

**Figure 1 jcm-09-00161-f001:**
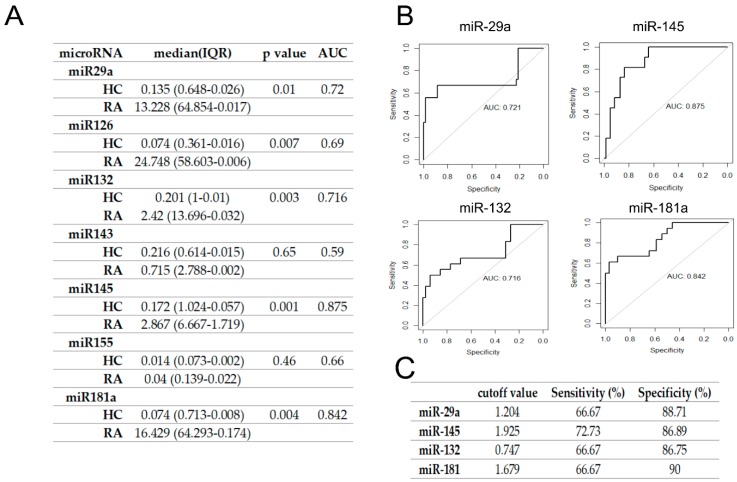
Serum miR- 29a, miR-145, miR-132, and miR-181a differentiated RA from HC. (**A**) Serum expression of seven miRNAs for a candidate RA biomarker. The area under curve (AUC) was calculated after plotting the receiver operating characteristic (ROC) curve. (**B**) ROC curve analyses of miR-29a, miR-145, miR-132, and miR-181a, which showed the highest values for AUC. (**C**) The sensitivity and specificity test for RA of miRNAs with the highest AUC value. The cutoff value was selected using a method that maximizes sensitivity and specificity simultaneously. IQR, Interquartile range; RA, rheumatoid arthritis; HC, healthy control.

**Figure 2 jcm-09-00161-f002:**
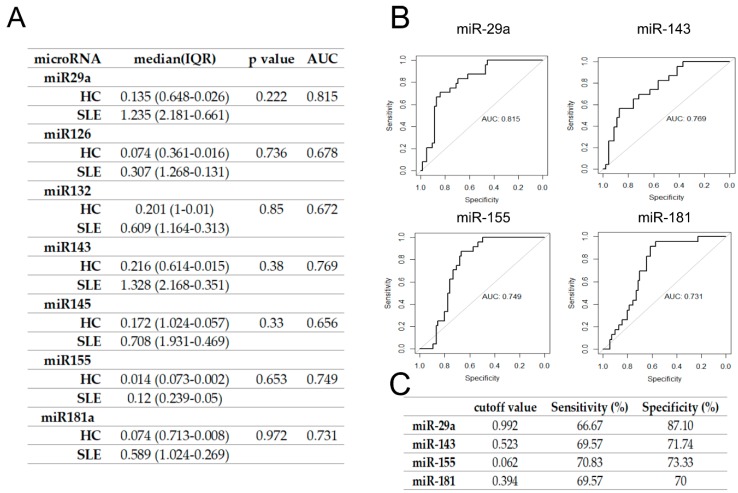
ROC curve analysis and calculated AUC values of seven miRNAs for the candidate SLE biomarker. (**A**) Serum expression of seven miRNAs for candidate SLE biomarker. The AUC was calculated after plotting the ROC curve. (**B**) ROC curve analyses of miR- 29a, miR-143, miR-155, and miR-181a, which showed the highest values for AUC. (**C**) The sensitivity and specificity test for SLE of miRNAs with the highest AUC value. Cutoff value was selected using a method that maximizes sensitivity and specificity simultaneously. SLE, systemic lupus erythematosus.

**Figure 3 jcm-09-00161-f003:**
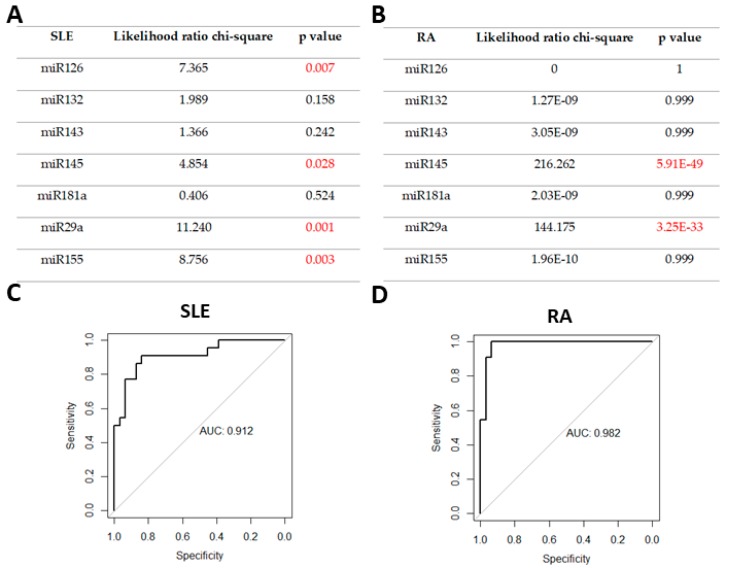
Identification of crucial miRNAs in (**A**) a SLE detection model; (**B**) a RA detection model; (**C**) ROC analysis of the combination of miR-126, miR-145, miR29a, and miR155 in SLE; (**D**) ROC analysis of the combination of miR-145 and miR-29a in RA.

**Figure 4 jcm-09-00161-f004:**
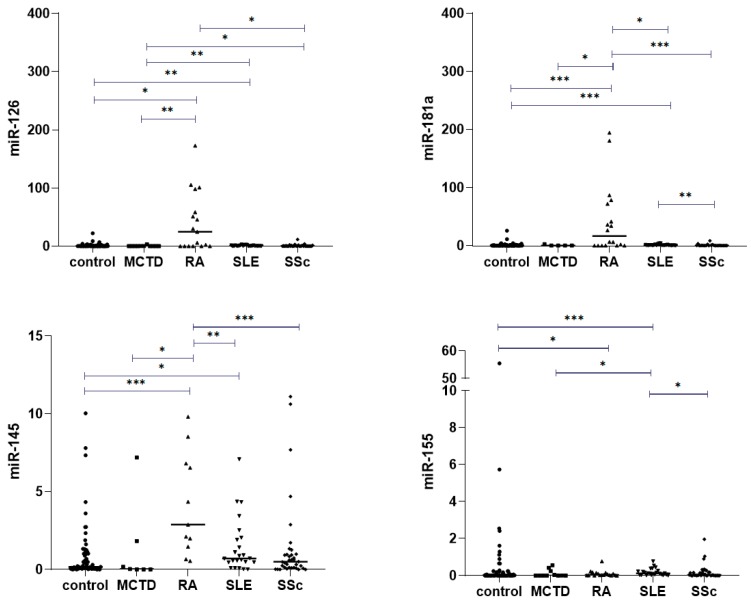
Serum expression of (**A**) miR-126, (**B**) miR-181a, (**C**) miR145, and (**D**) miR155 normalized to U6 between ACTD groups, determined by qRT-PCR. Box plots display numerical data through their quartiles. Only significant differences are indicated. *, *p* ≤ 0.05; **: *p* ≤ 0.01; *** *p* ≤ 0.001.

**Table 1 jcm-09-00161-t001:** Clinical description of patients who participated in the study.

Parameters	MCTD *n* = 15	SLE *n* = 24	RA *n* = 18	SSc *n* = 37
age	48.5 (58.75–33.25)	36 (46.5–32.75)	58 (68–49)	52.5 (57.5–41)
sex (female)	12 (85.71%)	19 (95%)	11 (64.7112%)	23 (63.89%)
disease duration (months)	126 (192–63)	72 (135–30.75)	54 (153–12)	
damage index	1 (3–0)	1 (1–0)	5.37 (5.76–4.2)	
CRP	5 (7–3)	4 (6–1.5)	32 (47.5–23.5)	2 (7–1)
ESR [mm/h]	10 (20–8)	11.7 (14.15–7.85)	45 (54–29)	11 (1–52)
**Autoantibody profile**				
anti–CCP	3 (23.08%)		11 (73.33%)	
RF	3 (25%)		8 (53.33%)	4 (13.79%)
anti-U1-RNP	10 (76.92%)	2 (12.5%)		
anti-A	8 (61.54%)	4 (25%)		
anti-C	5 (38.46%)	3 (18.75%)		
anti-70kD	9 (69.23%)			
ANA titer:				
<1280	4 (30%)	9 (37.5%)		10 (27%)
≥1280	9 (60%)	15 (62.5%)		27 (68%)
anti–dsDNA	1 (7.69%)	13 (81.25%)		0 (0%)
SSA – Ro52	2 (15.38%)	5 (31.25%)		7 (23.33%)
SSA – Ro60	0 (0%)	5 (31.25%)		1 (5.56%)
anti-SmB	4 (30.77%)	5 (31.25%)		0 (0%)
anti-SmD	0 (0%)	5 (31.25%)		0 (0%)
anti-Scl70	1 (7.69%)	0 (0%)		16 (57.14%)
Therapeutic profile	Azathioprine – 6%	Ciclosporin – 6%	Anti-IL–6%– 11%	Azathioprine – 13%
	Methotrexate – 23%	Azathioprine – 24%	Glucocorticoids – 61%	Cyclophosphamide – 7%
	Steroids(prednizon) – 77%	Cyclophosphamide – 10%	Anti-TNF – 39%	Other immunosuppressive drugs – 52%
	Chloroquine – 59%	Methotrexate – 10%	Methotrexate – 61%	methotrexate – 27%
		Steroids (prednizon) – 94%		Corticosteroids – 7%
		Chloroquine –52%		Vasodilators – 84%
		Hydroksychloroquine – 42%		

MCTD, mixed connective tissue disease; SLE, systemic lupus erythematosus; RA, rheumatoid arthritis; SSc, systemic sclerosis; RF, rheumatoid factor; anti-CCP, anti-cyclic citrullinated peptide autoantibodies, aCCP; anti-Scl-70, anti-topoisomerase I; anti-dsDNA, anti-double stranded DNA, CRP, *C-reactive protein*; ESR, *erythrocyte sedimentation rate*; SSA, Sjögren’s-syndrome-related antigen. continuous variables were presented as median and Interquartile range (IQR); Categorical variables were presented as percentages.

**Table 2 jcm-09-00161-t002:** Relative expression of circulating microRNA in ACTD patients compared to the healthy control group (expression in the control group is taken as 1).

**SLE**					
**Upregulated microRNA**	**Fold change**	***p*-value**	**Downregulated microRNA**	**Fold change**	***p*-value**
miR-29a	9.15	<0.0001			
miR-155	8.57	0.0002			
miR-143	6.15	0.0003			
miR-181a	7.959	0.0009			
miR-126	4.15	0.009			
miR-132	3.03	0.02			
miR-145	4.12	0.03			
**MCTD**					
**Upregulated microRNA**	**Fold change**	***p* value**	**Down regulated microRNA**	**Fold change**	***p* value**
miR-143	10.50	0.06	miR-126	4.35	0.09
miR-29a	8.98	0.1	miR-145	5.73	0.2
miR-132	7.08	0.3	miR-155	2.8	0.5
miR-181a	1.08	0.7			
**RA**					
**Upregulated microRNA**	**Fold change**	***p*-value**	**Downregulated microRNA**	**Fold change**	***p* value**
miR-181a	222.01	<0.0001			
miR-145	16.67	<0.0001			
miR-29a	97.99	0.004			
miR-132	12.04	0.007			
miR-126	334.43	0.01			
miR-155	2.86	0.03			
miR-143	3.31	0.2			
**SSc**					
**Upregulated microRNA**	**Fold change**	***p* value**	**Down regulated microRNA**	**Fold change**	***p* value**
miR-155	3	0.06	miR-181a	2.55	0.5
miR-126	2.68	0.2	miR-29a	1.12	0.7
miR-145	2.88	0.3	miR-143	2.45	0.9
miR-132	1.35	0.9			

Wilcoxon test, *p*-value < 0.05 was considered to be significant, marked by bold font.

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
