# Peer review of "The Serum Cell-Free microRNA Expression Profile in MCTD, SLE, SSc, and RA Patients"

_jcm, 2020, doi:10.3390/jcm9010161_

Round 1
Reviewer 1 Report
This study aims at determining whether miRNAs participating in the TLRs signaling pathway could serve as biomarkers differentiating MCTD or other ACTD entities from a healthy control group and between groups of patients.
I found some relevant aspects that must be revised and clarified:
PATIENTS AND METHODS
The table describing clinical and serological profiles of patients is complex and difficult to understand, so that in some diseases it includes lots of parameters while in others describes too low. Moreover, those clinical/serological parameters are not even related to the miRNAs analyzed. My advice is to perform a more simple and comprehensive table, eliminating data belonging exclusively to a disease (which could be included in the text if necessary). In the table only 2 SLE patients display Ab anti-histone positivity. However, on the results section their association with miRNA levels is highlighted. Do I have to interpret that a comparison has been made between 2 patients vs 22??? To evaluate circulating microRNAs, U6 is not appropriate for normalization. It should be used only for analysis performed on cells. Instead, the expression levels of plasma miRNAs should be normalized to the mean of sipked-in miRNA Cel-miR39. Other normalizations are also acceptable, including an internal normalization among the median levels of miRNAs evaluated.This point might explain the lack of coincidence among the data obtained by these authors and that of the majority of papers published in this matter.
RESULTS AND DISCUSSION
Results presented included four independent graphs that compare miRNA expression among pathologies. This representation does not allow to clearly distinguish expression levels among diseases. It would be interesting to make that comparison by performing heatmaps matching median levels of each miRNAs among the diseases evaluated. It would be also interesting to evaluate by ROC curve analyses if the seven identified circulating microRNAs (or some of them) could conform a signature (identified by lineal regression model analysis) that distinguish patients from healthy donors on each pathology, or even inside each autoimmune condition could identify any pathological aspect. It would be advisable to analyze if these miRNAs might influence the expression of key inflammatory mediators in those autoimmune patients (i.e. TNF, VEGF, IL-1, IL-6, IL-17, etc), for example by quantifying these molecules in serum and looking for correlations among their levels and those of the 7 miRNAs. The scarce correlations and associations identified by statistical analyses should be graphically represented. In the discussion section, the differences found among this study and those previously published in these diseases are extensively described. However, the potential physio pathological relevance of the miRNA alterations observed on each disease are not analyzed. No clear and significant conclusions are obtained from the study.Author Response
Thank you very much for the precious comments and suggestions that will undoubtedly improve the quality of the submitted manuscript.
Comment 1. The table describing clinical and serological profiles of patients is complex and difficult to understand, so that in some diseases it includes lots of parameters while in others describes too low. Moreover, those clinical/serological parameters are not even related to the miRNAs analyzed. My advice is to perform a more simple and comprehensive table, eliminating data belonging exclusively to a disease (which could be included in the text if necessary).
As Reviewer suggested we changed the Table 1. Please refer to the revised version of the manuscript.
Comment 2. In the table only 2 SLE patients display Ab anti-histone positivity. However, on the results section their association with miRNA levels is highlighted. Do I have to interpret that a comparison has been made between 2 patients vs 22???
Thank you very much for the Reviewer's comments. The sentence in the results section “In contrast, levels of miR-132 positively correlated with the presence of anti-histone antibodies (cor = 0.773, p = 0.001)” was delated. It was our oversight.
Comment 3. To evaluate circulating microRNAs, U6 is not appropriate for normalization. It should be used only for analysis performed on cells. Instead, the expression levels of plasma miRNAs should be normalized to the mean of sipked-in miRNA Cel-miR39. Other normalizations are also acceptable, including an internal normalization among the median levels of miRNAs evaluated.
This point might explain the lack of coincidence among the data obtained by these authors and that of the majority of papers published in this matter.
Evaluation of circulating microRNA in serum is really challenging and to obtain any results we had to optimize experiment starting from the isolation method to PCR. One of the standardization method was volume of sample and elution volume were always the same. In our laboratory we check the isolation and reverse transcription reaction testing samples with exogenous microRNA. During optimizing method, we test panel of endogenous reference genes. We follow this rules also in this study. However, in our experiment, among other references genes, U6 was the most stable, represented high Ct value in all studied samples and narrow and reproducible SD. This is the most important hallmarks for the endogenous control and consequently for reliable and reproducible results. Although U6 is not recommended as the best choice, particularly in cancer research, in the literature we can find also recommendation for U6. In 2015, Rice et. al in the paper entitled Housekeepig genes for studies of plasma microRNA: A need for more precise standardization stated that in plasma research U6 is one of the best (authors check panel of genes: Let-7a, Let-7d. Let-7g, miR-16, RNU48, miR-191, miR-223). Moreover we did not use cel-39 to normalize our results according to the literature among other written by McAlexander et al (2015). Authors in their paper about microRNA in cerebrospinal fluid stated that the spike-in [cel-miR-39] is useful for assessment of recovery but cannot be used for biological normalization (…) normalization by spike-in between experiments would not be inappropriate and was not done. In the picture below we present amplification plot of spike in which we use in our research. As it was mentioned above we evaluated the quality of isolation and reverse transcription carried out in our experiments by them, but due to the literature data given, we decided to use endogenous U6 to normalize expression.
Figure 1 Evaluation of sample quality (miRNA extraction and reverse transcription).
Comment 4. Results presented included four independent graphs that compare miRNA expression among pathologies. This representation does not allow to clearly distinguish expression levels among diseases. It would be interesting to make that comparison by performing heatmaps matching median levels of each miRNAs among the diseases evaluated.
We deeply appreciate the Reviewer’s professional remarks. We did the heatmaps at the first step of our work with these data, but in our opinion it doesn’t present obtained results well (please see figure below), that’s why we decided to prepare four independent figures, showing significant differences between ACTDs. We decided to put heatmap figure in supplementary file, as an addition to already made results presentation.
Comment 5. It would be also interesting to evaluate by ROC curve analyses if the seven identified circulating microRNAs (or some of them) could conform a signature (identified by lineal regression model analysis) that distinguish patients from healthy donors on each pathology, or even inside each autoimmune condition could identify any pathological aspect.
As reviewer suggested we performed ROC curve analysis evaluating diagnostic potential of microRNAs in case of SLE and RA patients. Using Likelihood ratio test for multivariate logistic regression model we have selected four microRNAs combination for SLE and two microRNAs combination for RA that increase diagnostic potential of these biomarkers. Please refer to the revised version of the manuscript.
Comment 6. It would be advisable to analyze if these miRNAs might influence the expression of key inflammatory mediators in those autoimmune patients (i.e. TNF, VEGF, IL-1, IL-6, IL-17, etc), for example by quantifying these molecules in serum and looking for correlations among their levels and those of the 7 miRNAs. The scarce correlations and associations identified by statistical analyses should be graphically represented.
We deeply appreciate the Reviewer’s professional remarks. Unfortunately, at this moment, we don’t have information on the key inflammatory mediators (i.e. TNF, VEGF, IL-1, IL-6, IL-17, etc) in all patients from our study group. Moreover, in this moment we are unable to do these analysis because we do not have a ELISA kites. But we agree with the reviewer that analysis protein level of cytokines such as TNF, VEGF, IL-1, IL-6, IL-17 in the context of miRNA is very important and we would like continue our research, that’s why we we've submitted another project of extenders, our subjects and also taking into account the protein level assessment - we are waiting for the results.
Comment 7. In the discussion section, the differences found among this study and those previously published in these diseases are extensively described. However, the potential physio pathological relevance of the miRNA alterations observed on each disease are not analyzed. No clear and significant conclusions are obtained from the study.
The potential physiopathological relevance of the miRNA alterations have been added to the discussion part.

Reviewer 2 Report
In the current study Stypinska and co-workers have analyzed expression levels of a number of circulating miRNAs in patients with different autoimmune connective tissue diseases and in healthy controls with the primary aim to identify miRNAs that would allow for distinction of MCTD patients.
The current manuscript is well readable in large parts, however, it may benefit from additional linguistic proof reading.
Major comments:
Abstract: Please spell out abbreviations (MCTD, ACTD, TLR etc.) when they are mentioned the first time.
Table 1: The clinical data presented here are extensive but also important. However, the content is hardly understandable in its present format. In order to increase readability the authors may consider using another format for presenting these data. Are all data on laboratory characteristics in patients with SLE needed? Also, ANA titers should be presented in a different way for increased clarity. Detailed data may be moved out to a supplementary file.
P.5, it is inappropriate to discuss the results or the reason for performing a certain analysis in the results part.
If results, as in case for differential expression analysis between MCTD and controls, are not significant, one cannot state first that specific miRNAs were up- or downregulated.
The part about SSc “Circulating miR-155, 29a, -143, -181a, -126, -132 and -145 in patients with SSc showed comparable levels with healthy subjects. Additionally, we observed that in SSc patients miR-155, miR-126, miR-145, miR-132 were upregulated, while miR-181a, miR-29a, miR-143 were downregulated.” is contradictory and needs to be explained and/or corrected.
In order to support clarity and interpretability of the data, I suggest to present the expression levels of the six/seven (it says six on page 5, while seven miRNAs are included in Table 2 on page 6) investigated miRNAs as a heat map. Table 2 as well the results part are rather confusing give the number of different comparisons between different diseases and between the different diseases and controls. Again, one cannot state that a miRNA is upregulated, when the result is not significant (p.6).
P.9, Again, please do not give non-significant results in a way as they were significant. It would be of interest for the reader to see plotted results of the correlation analyses. Which non-parametric test was used to test correlation?
The discussion part needs to be revised in terms of shortening and focusing on the important aspects in order be increase readability. Some parts of the discussion section were already mentioned lengthily in the introduction section.
The negative results of the differential expression analysis between MCTD patients and healthy controls, while at the same time differences were observed between MCTC and patients with other ACTDs, need to be discussed further. Are MCTD patients (or at least their miRNa expression) in general more similar to healthy controls? Are the investigated miRNA the most relevant ones for MCTD pathology or may other miRNAs be included instead?
Inconclusive or contradicting results of miRNA expression in SLE and RA (in comparison with healthy controls and also between clinical subgroups) need to be presented more clearly in those aspects where the results of the current study did not replicate previous findings.
Minor comments:
Figure 1 to 4: The label of the y-axis needs to be changed in order to support understandability. Also, the authors may consider to combine all four figures into one figure with A), B), C), D) subparts, as they are in their current format unnecessarily large and redundant (e.g. in terms of sample group legends, figure legends and p-value explanations). The table part in each figure is more confusing than informative, and information about number of samples in each group could be incorporated easily into the main plot; all other data on mean, sd, median, min and max can be deducted from the plots themselves.
P.10, what is the rational for underlining text and using bold text?
Author Response
Thank you very much for the precious comments and suggestions that will undoubtedly improve the quality of the submitted manuscript.
Major comments:
Comment 1. Abstract: Please spell out abbreviations (MCTD, ACTD, TLR etc.) when they are mentioned the first time.
As the Reviewer’s suggested we added in the abstract and in the main text the define all abbreviations the first time was used. Please refer to the revised version of the manuscript.
Comment 2. Table 1: The clinical data presented here are extensive but also important. However, the content is hardly understandable in its present format. In order to increase readability the authors may consider using another format for presenting these data. Are all data on laboratory characteristics in patients with SLE needed? Also, ANA titers should be presented in a different way for increased clarity. Detailed data may be moved out to a supplementary file.
As Reviewer suggested we changed the Table 1. Please refer to the revised version of the manuscript.
Comment 3. P.5, it is inappropriate to discuss the results or the reason for performing a certain analysis in the results part.
According to suggestion it has been reedited.
Comment 4. If results, as in case for differential expression analysis between MCTD and controls, are not significant, one cannot state first that specific miRNAs were up- or downregulated.
We accepted these suggestions and made the appropriate changes, please refer to the revised manuscript.
Comment 5. The part about SSc “Circulating miR-155, 29a, -143, -181a, -126, -132 and -145 in patients with SSc showed comparable levels with healthy subjects. Additionally, we observed that in SSc patients miR-155, miR-126, miR-145, miR-132 were upregulated, while miR-181a, miR-29a, miR-143 were downregulated.” is contradictory and needs to be explained and/or corrected.
Thank you very much for the Reviewer's kind comments. We corrected this information. Please refer to the revised version of the manuscript “We observed that in SSc patients miR-155, miR-126, miR-145, miR-132 were upregulated, while miR-181a, miR-29a, miR-143 were downregulated, however this differences was not significant”.
Comment 6. In order to support clarity and interpretability of the data, I suggest to present the expression levels of the six/seven (it says six on page 5, while seven miRNAs are included in Table 2 on page 6) investigated miRNAs as a heat map. Table 2 as well the results part are rather confusing give the number of different comparisons between different diseases and between the different diseases and controls. Again, one cannot state that a miRNA is upregulated, when the result is not significant (p.6).
We deeply appreciate the Reviewer’s professional remarks. We did the heatmaps at the first step of our work with these data, but in our opinion it doesn’t present obtained results well (please see figure below), that’s why we decided to prepare four independent figures, showing significant differences between ACTDs. We decided to put heatmap figure in supplementary file, as an addition to already made results presentation.
Comment 7. P.9, Again, please do not give non-significant results in a way as they were significant. It would be of interest for the reader to see plotted results of the correlation analyses. Which non-parametric test was used to test correlation?
Spearman test has been used to check the correlation.
Comment 8. The discussion part needs to be revised in terms of shortening and focusing on the important aspects in order be increase readability. Some parts of the discussion section were already mentioned lengthily in the introduction section.
As Reviewer suggested we re-edited the Discussion section. Please refer to the revised version of the manuscript.
Comment 9. The negative results of the differential expression analysis between MCTD patients and healthy controls, while at the same time differences were observed between MCTC and patients with other ACTDs, need to be discussed further. Are MCTD patients (or at least their miRNa expression) in general more similar to healthy controls? Are the investigated miRNA the most relevant ones for MCTD pathology or may other miRNAs be included instead?
We deeply appreciate the Reviewer’s professional remarks. As Reviewer suggested we added some information about this in the Discussion section. Please refer to the revised version of the manuscript.
Comment 10. Inconclusive or contradicting results of miRNA expression in SLE and RA (in comparison with healthy controls and also between clinical subgroups) need to be presented more clearly in those aspects where the results of the current study did not replicate previous findings.
As the Reviewer’s suggested we add some information in the Discussion section.
Minor comments:
Comment 11. Figure 1 to 4: The label of the y-axis needs to be changed in order to support understandability. Also, the authors may consider to combine all four figures into one figure with A), B), C), D) subparts, as they are in their current format unnecessarily large and redundant (e.g. in terms of sample group legends, figure legends and p-value explanations). The table part in each figure is more confusing than informative, and information about number of samples in each group could be incorporated easily into the main plot; all other data on mean, sd, median, min and max can be deducted from the plots themselves.
The figures have been corrected. We hope that this form of data presentation will be satisfactory for you.
Comment 12. P.10, what is the rational for underlining text and using bold text?
The underlining was not intentional, it has been removed from the text. Thank you for noticing this error.

Reviewer 3 Report
Autoimmune diseases like SLE, RA and mixed connective tissue disease, are mostly multifactorial and multi symptomatic. Due to the varied yet overlapping pathogenesis, the identification or classification of these diseases, especially early on is difficult. In that regard, in this paper the authors have attempted to analyze and compare cohorts of SLE, RA, SS, MCTD etc. to understand the role of miRNAs in MCTD, and identify potential biomarkers.
The study is clearly presented and well written. The study would be helped by answering the following questions and concerns.
The authors were unfortunately unable to identify a stand out miRNA specifically over expressed in MCTD. However miRNA 126 was at its lowest expression similar to healthy controls. The limitation could be due to the limited number of patient blood samples and/or due to the limited number of miRNAs scanned (previously implicated in TLR-IFN related pathways). However, the authors have not stated in the clinical description if the patients were on therapeutics and the details of them. A clear indication of the therapy the various patients were on would be necessary and will help future studies. Although a table has been provided, in the figures, the Y axis could be presented in two segments to enhance the values between 0-1 so as to highlight the differences between groups. As currently presented only the outliers are determining the way the data has been presented. The authors state in results "Moreover, our analysis showed significant association between examined miRNAs and autoantibodies presence in SLE patients. 199 Levels of miR-145 (cor= -0.579, p = 0.02) and miR-155 (cor= -0.608, p = 0.01) negatively correlated 200 with the presence of anti-dsDNA antibodies." This is a significant observation but seems lost in the results as a statement. The authors should present the data as a figure, especially as that was not the case for other diseases like RA and SSc.
Author Response
Thank you very much for the precious comments and suggestions that will undoubtedly improve the quality of the submitted manuscript.
Comment 1. The authors were unfortunately unable to identify a stand out miRNA specifically over expressed in MCTD. However miRNA 126 was at its lowest expression similar to healthy controls. The limitation could be due to the limited number of patient blood samples and/or due to the limited number of miRNAs scanned (previously implicated in TLR-IFN related pathways).
Thank you very much for the Reviewer’s kind comments. We agree with Reviewer conclusion. Our study sample size is relatively small; however, MCTD is a very rare disease with a prevalence of about 10 cases/100,000 (for example rheumatoid arthritis (RA) affecting 1/100 people). We did not choose miRNAs based on microarray/NGS studies and thus our analyses were limited to individually measured miRNAs in this study. We believe that our result could serve as a reference for other investigators or be used in a bigger statistic meta-analysis and contribute for making clear conclusions. As Reviewer suggested we added information about group size in limitation of our study. Please refer to the revised version of the manuscript.
Comment 2. However, the authors have not stated in the clinical description if the patients were on therapeutics and the details of them. A clear indication of the therapy the various patients were on would be necessary and will help future studies.
We deeply appreciate the Reviewer’s professional remarks. In Table 1 we added information about therapeutic profile. Further research is needed to clarify the multitude of effects that miRNAs influence in order to utilize miRNAs as diagnostic tools and disease modifying therapies in MCTD, SLE, RA and SSc patients.
Comment 3. Although a table has been provided, in the figures, the Y axis could be presented in two segments to enhance the values between 0-1 so as to highlight the differences between groups. As currently presented only the outliers are determining the way the data has been presented.
According to the suggestion figures has been changed.
Comment 4. The authors state in results "Moreover, our analysis showed significant association between examined miRNAs and autoantibodies presence in SLE patients. 199 Levels of miR-145 (cor= -0.579, p = 0.02) and miR-155 (cor= -0.608, p = 0.01) negatively correlated 200 with the presence of anti-dsDNA antibodies." This is a significant observation but seems lost in the results as a statement. The authors should present the data as a figure, especially as that was not the case for other diseases like RA and SSc.
We deeply appreciate the Reviewer’s professional remarks. As Reviewer suggested we presented this data as a Supplementary figure 1 in supplementary files.

Round 2
Reviewer 1 Report
The authors have adequately answered to all my concerns.